# Influence of Type of Filler and Bitumen on the Mechanical Performance of Asphalt Mortars

**DOI:** 10.3390/ma15093307

**Published:** 2022-05-05

**Authors:** Raul Tauste-Martínez, Ana Elena Hidalgo, Gema García-Travé, Fernando Moreno-Navarro, María del Carmen Rubio-Gámez

**Affiliations:** 1E.T.S. de Ingeniería de Caminos, Canales y Puertos, University of Granada, C/Severo Ochoa S/N, 18071 Granada, Spain; rtauste@ugr.es (R.T.-M.); labic@ugr.es (G.G.-T.); fmoreno@ugr.es (F.M.-N.); 2Laboratorio Nacional de Materiales y Modelos Estructurales (LANAMME), University of Costa Rica, San Jose 2060, Costa Rica; anaelena.hidalgo@ucr.ac.cr

**Keywords:** fine aggregate matrix, filler, polymer-modified bitumen, asphalt mortar, asphalt mixture, fatigue performance

## Abstract

This article presents a new methodology of analysis based on a fast-running experimental procedure to characterise the mechanical response of asphalt mortars in terms of stiffness, ductility, and fatigue resistance. This was achieved using the DMA (Dynamic Mechanical Analyser) three-point bending configuration. The study was carried out by considering the employment of different types of fillers such cement and CaCO_3_ and different types of binders such as conventional asphalt binder (B35/50) or modifided polymer-modified bitumen (PMB 25/55–65). From the results of this study, the filler was found to have a greater influence on the stiffness and ductility of the asphalt material, while bitumen had a higher effect on the fatigue life of the asphalt mortar. Fatigue life was observed to increase with the use of a polymer-modified binder, while a lower degree of permanent deformation and higher bearing capacity achieved by the use of cement instead of calcium carbonate as active fillers.

## 1. Introduction

The continual increases in traffic levels, coupled with greater environmental burdens such as more extreme climatic events due to climate change, have led to the need to design asphalt mixtures with a higher performance level [1]. Moreover, current investments in road networks in developed countries have been evolving toward the preservation of existing infrastructures rather than the construction of new ones [2]. This has led to a need to optimise existing resources in order to achieve solutions that can improve the performance of roads and to guarantee their future durability. To achieve this, it is fundamental to find new tools that can help to improve the design of asphalt mixtures. This should be accomplished in a manner in which designs could be more agile, the use of new materials could still be explored, and the limited resources available can be optimised.

An adequate selection of the materials and asphalt mixtures used in the construction of road structures exerts a fundamental role in the performance, durability, and functionality of the transport infrastructure. An optimal asphalt mixture design would guarantee sufficient bearing capacity, enough flexibility to counteract the initiation of cracking at low stress levels [3], a good interaction between the components [4], and the capacity of resist fatigue associated with both traffic loads and thermal stresses [5]. Fatigue phenomena could give way to the progressive cracking and breakage of the asphalt mixture layers of the pavement structure, which is one of the main reasons behind structural failure in pavements [6]. There are several tests for evaluating the fatigue resistance at the asphalt mixture level; however, alternative methods could be useful in order to explore the properties of material via fewer testing steps. In turn, this would also provide a quicker understanding of the material needing lower investment costs without compromising the representativeness of the mechanical behaviour described.

In this sense and in spite of its relative low percentage among the asphalt mixture, the materials conforming the Fine Aggregates Matrix (FAM) (i.e., bitumen, the finest aggregate fraction of asphalt mixture and filler) are responsible for providing appropriate granular material cohesion [7]. FAM (or asphalt mortar) also conditions the viscoelastic response of the entire mixture and, with it, its resistance to fatigue and cracking [8]. In addition, both the filler and binder are, despite its low proportion, the more expensive constituent materials in the mixture (especially if active fillers or polymer-modified bitumen are employed). Therefore, it is important to know, as accurately as possible, the actual influence of the employed asphalt binder (penetration or modified) and filler type (natural or active) on the mechanical response of the mixtures [9].

Different studies have analysed the basic physical-chemical characteristics of the filler or rheologic performance at asphalt binder level [10,11]. Nevertheless, this kind of analyses lack the consideration of the interaction between asphalt, filler, and the finest part of the aggregates as a fundamental factor conditioning the mechanical behaviour of asphalt materials. For this reason, the analysis of an intermediate level such as FAM characterization could be helpful in precisely determining the repercussions of using different kinds of fillers and bitumens in phenomena such as fatigue cracking. This is possible by studying it at the scale that most conditions the viscoelastic response of the material.

There are several studies [12,13,14] that have employed this kind of analysis as a method of characterizing the fatigue resistance of asphalt mixtures. These methods extend the principle used in the characterization of bitumen rheology and make use of samples that are usually cylindrical, which are tested under torsion. Following this, different oscillatory sweep tests (strain sweep, time sweep, relaxation, linear amplified sweep (LAS), creep, and recovery, etc.) are carried out at different frequencies and temperatures to characterise the viscoelastic properties of the material, its recovery capacity, and fatigue resistance [15]. According to their results, the analysis of asphalt mortars has been shown to be a viable method since it deals with the problem of the heterogeneous composition of asphalt mixtures through the characterization of a more homogeneous material (i.e., FAM) [16,17]. This makes it possible to examine the precise influence of different fillers or even polymer-modified binders, since it enables the analysis of the maximum specific surface of asphalt mixtures. At this scale, a higher concentration of binder is anticipated and a sensitivity closer to the entire mixture can be achieved [18,19]. Nonetheless, despite this, there is still a certain lack of a common methodology regarding fatigue characterization (with different tests and failure criterion considered) that can provide confidence in this type of analysis. In turn, this limits the spread of its use as a tool in the design for asphalt mixtures and the evaluation of different materials related to their mastics.

Therefore, the aim of this study was to analyse the influence of the type of filler and bitumen on the stiffness, ductility, and fatigue resistance. This was achieved through the implementation of a methodology of analysis based on 3 point-bending DMA (Dynamic Mechanical Analyser) fatigue characterization for asphalt mortars (FAM).

## 2. Materials and Methods

### 2.1. Materials

Four asphalt mortars were studied (Table 1) to establish the effect of the type of filler and bitumen on their mechanical response. All of them were designed based on the properties of the materials employed in BBTM mixtures (EN 13108-2) [20]. This mixture type (commonly used at higher traffic levels) requires the use of active fillers and polymer-modified bitumen being the mortar fraction, which is a key role on its mechanical response.

To establish the influence of the filler employed, two mortars were manufactured using Portland cement as an active filler, and the other two were manufactured using calcium carbonate (CaCO_3_). Similarly, two mortars were manufactured using a conventional penetration bitumen (B35/50) and the other two mortars were manufactured using a polymer-modified bitumen (PMB 25/55–65) to analyse the influence of this type of bitumen (Figure 1).

Mortars were composed of 64.4% of limestone sand (washed to eliminate any filler particles), 27.6% of filler, and 8.0% of bitumen over the total asphalt mortar weight. The amount of binder corresponding to the mortar fraction was calculated according to the methodology employed in other studies [7,14,19]. The main characteristics of these materials are summarised in Table 2, Table 3 and Table 4.

The mixing process between the different components was carried out at 165 °C of temperature. After that, cylindrical specimens with a diameter size of 150 mm were compacted using a gyratory compactor (EN 12697-31) [27] at a temperature 160 °C until reaching a void content near to 0% (which gives way to specimen heights round 40 mm). By reducing to the maximum the air voids present in the material, the adhesiveness between aggregates and binder can be analysed more precisely [14]. Afterward, specimens were sawn using a precision coordinate cutting saw to obtain 8.5 × 8.5 × 50 mm prismatic samples. These would be used in the DMA (Dynamic Mechanical Analyser) characterization without the need to be coupled to the device as in other FAM characterization torsion tests. Figure 2 shows the aspect of the specimens manufactured and Table 5 shows the density of the different mortars manufactured.

### 2.2. Testing Plan

This investigation makes use of a new methodology of analysis that allow the assessment of the influence of using different types of active filler and asphalt binder in the mechanical performance of asphalt materials. By applying this, it is possible not only to determine the basic characteristics (such as bearing capacity, flexibility, or fatigue resistance) that condition the mechanical performance of asphalt mortars or mixtures, but we can also establish the relation between them and the changes induced by changes in the asphalt mortar’s composition. This will enhance the understanding of the influence of different fillers and bitumen types in the mortars helping in the improvement of the design of asphalt materials.

In order to achieve this, a three-point bending test (Figure 3) was performed with DMA Discovery HR 30™ equipment from TA Instruments™ (New Castle, DE, USA) on the prismatic specimens previously referred to. The use of this type of configuration instead of the most commonly used torsional one [12,17,18] allows bringing the material characterization closer to the methods commonly used in fatigue characterization at the level of mixtures such as UGR-FACT [29], 3-point fatigue test [30], and 4 point-bending test [31], among others. This new test allows the assessment of ductility, stiffness, and the fatigue response of the mortar, enhancing the comprehension about the influence of the finest fraction of aggregates and mastic components on the mechanical resistance of the material [19].

Based on these considerations, for each type of mortar, 9 specimens were tested divided into 3 sub-groups of 3 samples. The test of each sub-group was performed until reaching fatigue cracking failure at 3 different strain amplitudes: 40, 60, and 80 μm (DMA in strain-controlled mode). The load (of sinusoidal character) was applied at a frequency of 5 Hz and at a temperature of 35 °C. This temperature allows obtaining adequate test durations without compromising the adequate characterization of the material. In addition, the material is tested with demanding conditions in relation to fatigue response. The different values obtained were then used to define fatigue laws that helps in understanding the behaviour of the material against this phenomenon in different load conditions.

To establish the failure cycle, this methodology does not make use of a reduction in modulus value (usually between 40 and 60%), the definition of failure from the maximum phase angle value, or the establishment of different indices based on viscoelastic models as used in other studies [13,16]. Instead, image control was employed during all tests by making use of the live camera integrated in the equipment. The appearance of macro-cracks in the specimens along with the control of the evolution of the maximum force registered in each load cycle (a sharp drop in this force indicated the apparition of the macro-crack) makes it possible to examine the failure cycle (Figure 4). Alternatively, a maximum number of 250,000 cycles was established as failure criteria in cases where cracking did not fully develop beforehand.

This test allows the assessment of ductility, bearing capacity, and fatigue response of the material. These parameters are related to the performance of the FAM components within the asphalt mixture and condition and its durability. In this fashion, the mechanical behaviour of asphalt mortars can be studied by using the different parameters derived from the analysis of the three point-bending test [14]:

Stiffness (S). This first parameter evaluates the bearing capacity of the mortar, a key element in ensuring that the mixture has adequate resistance to permanent deformations [32]. It is defined as the relation between the force applied and the deflection induced in the mortar specimen after 1000 cycles. Stiffness uses the Young’s Modulus obtained from Equation (1) [33], and it is a measurement of the visco-elastic behaviour of the mortar (as it becomes higher, the mortar behaviour becomes more elastic). This is possible because all tests share same temperature and load conditions (amplitude and frequency):
(1)$$S=\frac{F\cdot l^{3}}{4\cdot d\cdot b\cdot h^{3}}$$where *F* is the maximum force applied in the load cycle 1000 (at this cycle, the sample is considered undamaged and within the viscoelastic range); *l* is the distance between the supporters; *d* is the deflection displacement measured in the cycle 1000; *b* is the width of the sample; and *h* is the sample height.Maximum deflection (dmax). It is defined as the accumulated deflection in the specimen before the macro-crack appearance (Figure 5). This parameter measures the ductility (ability to deform before cracking) of the material analysed, a property related to fatigue resistance, which is particularly important at low temperatures [34]. As the maximum deflection becomes higher, mortar ductility also becomes higher.Fatigue life (Nf). This parameter is defined by the total number of cycles applied in the sample before the appearance of a macro-crack. The parameter is a direct measurement of the fatigue resistance of the mortars (as this parameter increases, the resistance of the material also increases).

## 3. Results and Discussion

Based on the methodology previously described, the results presented enable the characterization of the influence of both the type of filler and binder on the behaviour of asphalt mortars.

### 3.1. Stiffness (S) and Ductility (dmax)

The following figures present the values of Stiffness and Maximum deflection (previously defined) obtained from the nine samples tested for each mortar at different levels of load. The representation of these values allows obtaining a general idea of the mechanical behaviour of the material and the differences that occur due to changes in its composition in terms of bearing capacity and flexibility.

Figure 6 and Figure 7 present the stiffness and deflection results for the Portland cement and calcium carbonate active fillers, respectively. The two binders are also represented in these figures (both the penetration or polymer-modified bitumen). The representation of these two parameters makes it possible to differentiate the mechanical response of the asphalt mortars manufactured with different active filler materials.

In terms of stiffness, mortars made with the cement filler (Figure 6) had values between 300 and 400 MPa, which were slightly higher in the case of the M3 mortar (made with the polymer-modified binder). In the case of calcium carbonate filler (Figure 7), a similar behaviour was observed for both types of binder. However, the bearing capacity was found to be lower with values between 150 and 250 MPa, which is always slightly higher for the mortar manufactured with modified bitumen (PMB 25/55–65).

Regarding ductility, despite the type of bitumen employed, it becomes clear that the mortars manufactured with CaCO_3_ experienced a higher value of deflection. Therefore, it is more flexible than those that incorporate cement as an active filler. The use of polymer-modified bitumen tends to slightly reduce the flexibility of the material following a similar trend to what was observed for stiffness.

According to these results, both the stiffness and ductility proved to be sensitive to the filler and bitumen typologies employed. It was also possible to demonstrate the utility of this kind of analysis to evaluate their influence in the bearing capacity and the resistance of the asphalt mortars to both cracking and permanent deformations. In summary, mortars manufactured with cement filler (Mortars M1 and M3) were found to be more rigid and less ductile compared to mortar specimens made with CaCO_3_ filler (M2 and M4). The specimens manufactured with conventional binder (B35/50, M1 and M2) were more flexible and less rigid than the ones made using a polymer-modified binder (M3 and M4).

### 3.2. Fatigue Life (Nf)

In this section, the fatigue life (Nf) values obtained in the different mortars for each loading condition were used to construct fatigue laws. For each specimen, the applied load value (corresponding to the required deformation level) is plotted against the number of cycles that was resisted until macro-crack appearance. A potential regression of these values allows obtaining a law describing the fatigue behaviour of the material against different levels of solicitation.

Figure 8 shows the fatigue laws of those asphalt mortars manufactured with the same type of filler. In this manner, Figure 8a displays the results of mortars with cement as active filler (M1 and M3), while Figure 8b presents those fabricated with CaCO_3_ (M2 and M4). This makes it possible to compare the influence of the asphalt binder on the fatigue response of these materials.

The differences found in fatigue resistance for the cement mortars (Figure 8a) show that the mortar manufactured using a B 35/50 penetration bitumen (M1) has a significantly lower (around a 500% less) fatigue life than the mortar that employed polymer-modified bitumen (M3).

Comparing the mortars that used CaCO_3_ as a filler (Figure 8b), the differences between the two asphalt binder typologies was slightly less significant but still visible. The mortar that used the modified bitumen (M4) was found to have a higher fatigue resistance than the one employing a conventional penetration binder (M2).

These results show that this kind of analysis is sensitive enough to distinguish the effect of different kinds of asphalt binders when using a determined kind of filler in asphalt mortar formulation. In both cases, it becomes clear that the use of polymer-modified bitumen helps to extend the fatigue life of these mortars. This is in accordance with other studies [19,35,36] and with the previous results regarding increased stiffness and lower deflection (Figure 6 and Figure 7). These are indicative of the increased cohesion between the asphalt mortar components that contribute to enhancing fatigue resistance in mortars.

Similarly, Figure 9 analyses the influence of the filler type on the fatigue response of asphalt mortars, comparing the fatigue laws of the mortars manufactured with the same kind of asphalt binder: B 35/50 (M1 and M2, Figure 9a) or PMB 25/55–65 (M3 and M4, Figure 9b).

Considering that, as seen previously, the mortars using polymer-modified bitumen in its formulation reached a higher fatigue life. Here, it can be observed that, regardless the kind of bitumen used, the mortars manufactured with cement (M1 and M3) have higher fatigue resistance than those that employed CaCO_3_ as active filler (M2 and M4). The differences between the fatigue life of the mortars manufactured with different types of filler becomes higher as the load amplitude tested is lower. This could indicate that beyond a certain level of load, the influence of the typology of bitumen would be more significant than the type of filler in relation with fatigue resistance. These results are again in accordance with those found in Figure 6 and Figure 7. This is particularly true when considering that a higher degree of stiffness (such as the one obtained through the use of cement) leads to an extended fatigue life, as it was also pointed out by previous studies [14,37].

Finally, in order to better distinguish the influence of the of filler and bitumen typologies, Figure 10 presents the relative maximum deflection (dmax) or ductility, stiffness (S), and fatigue life (Nf) together. For each one of these parameters, a ranking between the different mortars was established by assigning a value on the scale 1 to 4 to each one (where 1 is the lowest value and 4 is the largest value). Then, the values were represented in the graph. The ranking is settled according to the average value of each parameter defined at the load of 10 N, as observed in Table 6.

From this figure, it is possible to observe that the design component that most contributes to a higher fatigue life (Nf) is the typology of bitumen employed since mortars manufactured with polymer-modified bitumen (M3 and M4) are the ones with larger fatigue resistance. Regarding the other two parameters, ductility (dmax) and stiffness (S), these manifested an inverse result. Specifically, the higher the rigidity, the lower the maximum deflection. Regardless of this, both parameters were found to be more affected by the type of filler used in the design of the asphalt mixture. Mortars with cement as active filler (M1 and M3) were more rigid than those manufactured using CaCO_3_ (M2 and M4). Therefore, the former would have a higher bearing capacity and permanent deformation resistance, whereas the latter would present a higher ductility and would be more flexible.

## 4. Conclusions

This study centers on the application of a new methodology of analysis of asphalt mortars based on 3-point bending fatigue tests performed with DMA equipment. This was used to establish the influence of different types of active filler and asphalt binder on their mechanical performance. According to the results showed, some conclusions can be reached:The methodology presented made it possible to identify the effect of the filler on the stiffness and ductility of asphalt mortars, both being parameters that condition the mechanical performance of asphalt mixtures. In this case, it can be observed how mortars manufactured with Portland cement are more rigid and have lower deflection (dmax) in comparison with CaCO_3_ mortars.These two parameters are also affected by the bitumen type employed. It was possible to distinguish a slightly more rigid and less flexible behaviour for the polymer-modified binder regardless of the filler type employed.The highest fatigue resistance was achieved through the employment of polymer-modified binder regardless of the filler used. Nonetheless, when the same kind of bitumen (either penetration or modified) was employed, the higher degree of stiffness of cement mortars helps achieve larger fatigue lives than mortars manufactured with CaCO_3_.It was possible to establish that the typology of bitumen used will affect, to a greater extend, the fatigue life of asphalt materials. Meanwhile the stiffness, ductility, and the properties that depend on them (bearing capacity, permanent deformation resistance, flexibility, etc.) will be conditioned more by the kind of filler employed.

According to the results found, it can be concluded that using asphalts mortars and the test protocol based on DMA 3-point bending configuration makes it possible to assess the influence of different types of filler and bitumen on the mechanical performance of these materials. This will help simplify its design and characterization in a manner that requires fewer resources.

## Figures and Tables

**Figure 1 materials-15-03307-f001:**
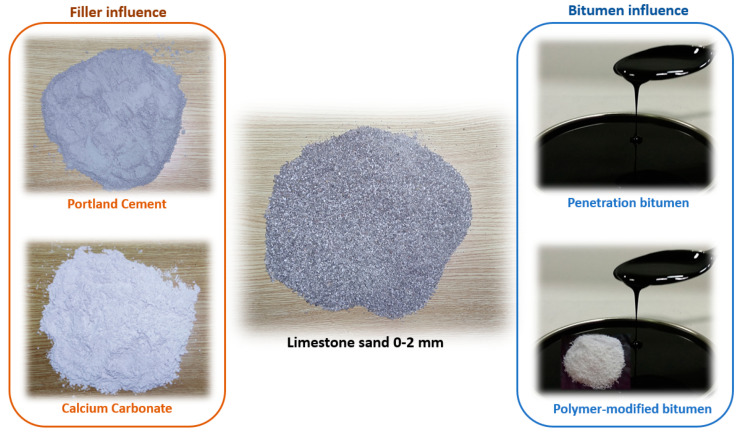
Components of the asphalt mortars studied.

**Figure 2 materials-15-03307-f002:**
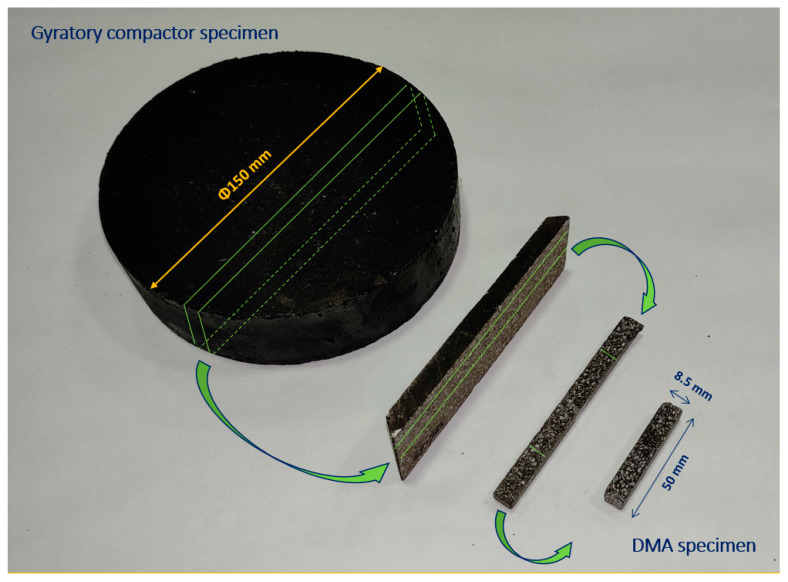
Obtaining process of the mortar samples used in DMA tests.

**Figure 3 materials-15-03307-f003:**
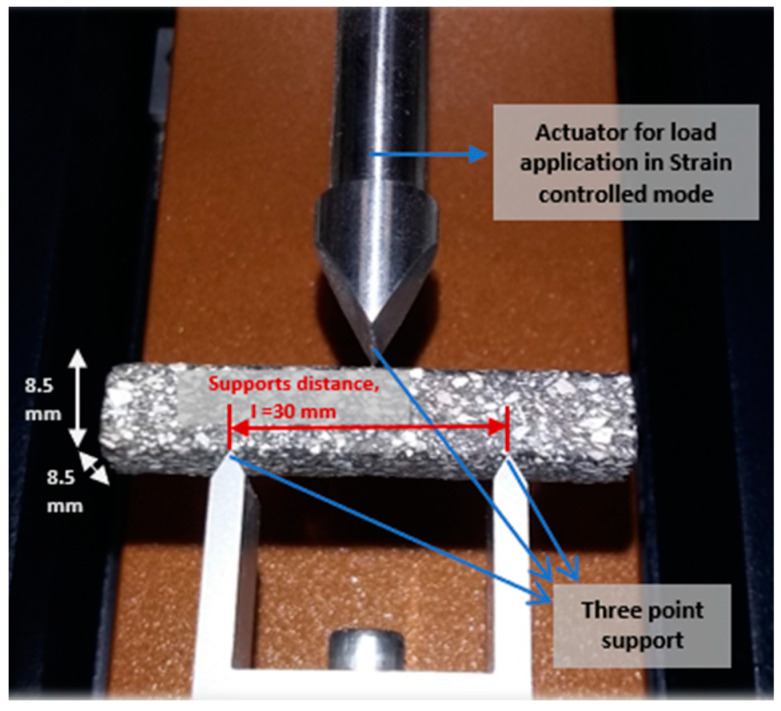
Layout of the three-point bending configuration employed in mortar characterization.

**Figure 4 materials-15-03307-f004:**
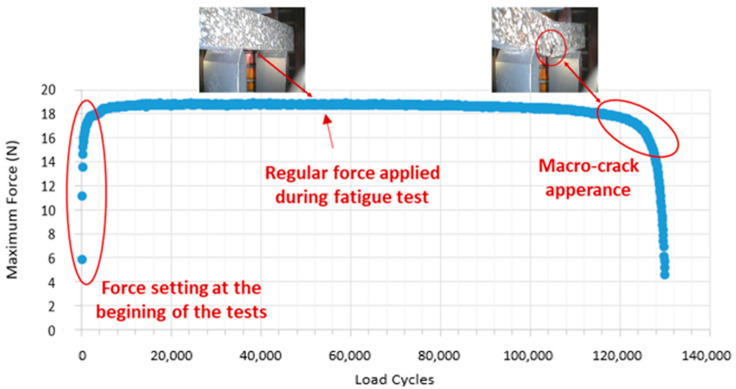
Example of the maximum force development and image monitoring in the fatigue 3-point bending test in DMA.

**Figure 5 materials-15-03307-f005:**
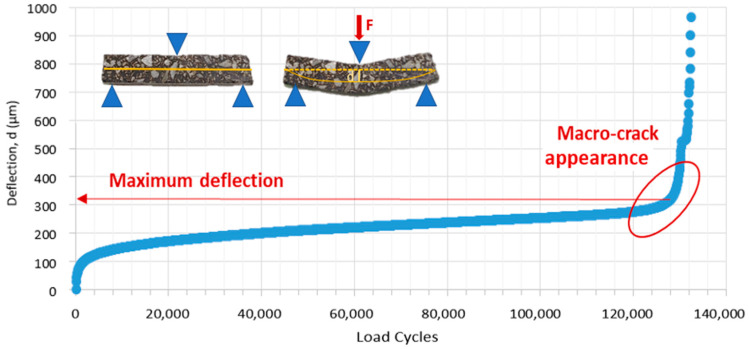
Deflection (d) registered during 3-point bending fatigue tests.

**Figure 6 materials-15-03307-f006:**
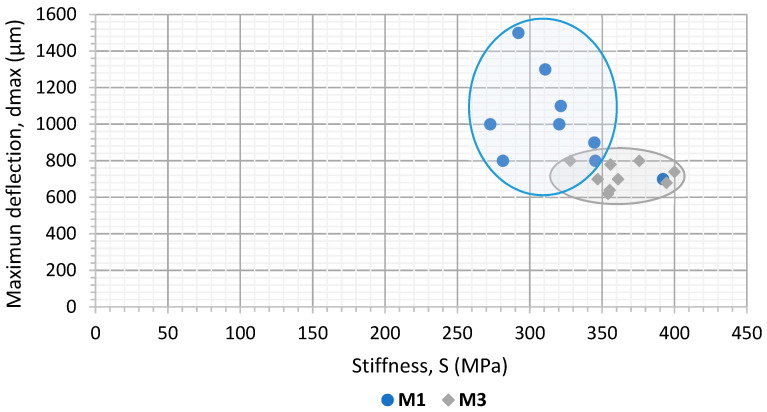
Results of stiffness and maximum deflection measured the mortars with cement active filler. M1 conventional binder M3 modified binder.

**Figure 7 materials-15-03307-f007:**
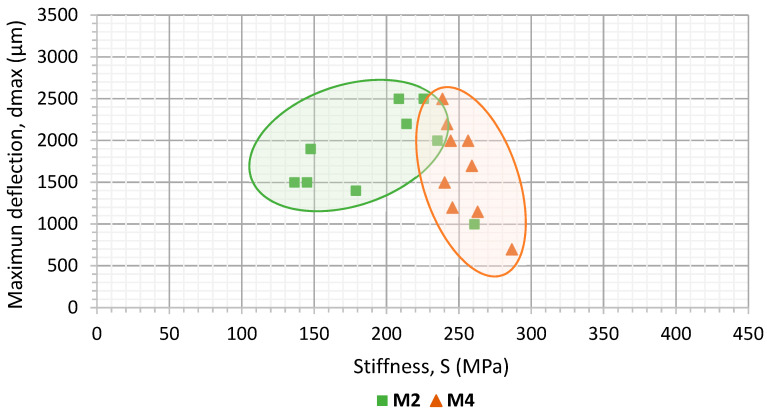
Results of stiffness and maximum deflection measured the mortars with CaCO_3_ active filler. M2 conventional binder M4 modified binder.

**Figure 8 materials-15-03307-f008:**
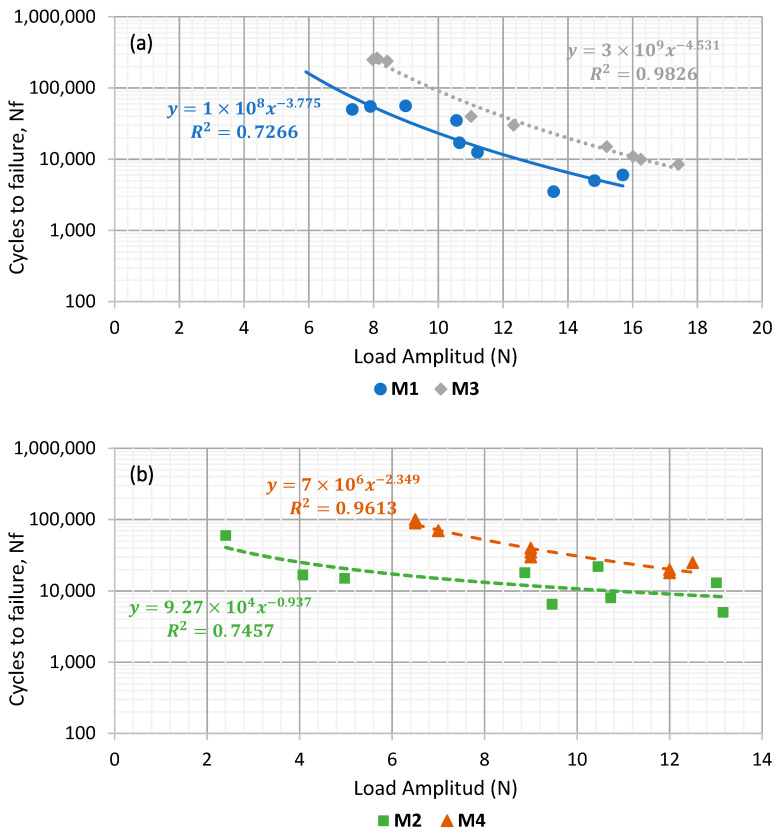
Comparison between types of bitumen for mortars manufactured with Portland cement (**a**) and CaCO_3_ (**b**).

**Figure 9 materials-15-03307-f009:**
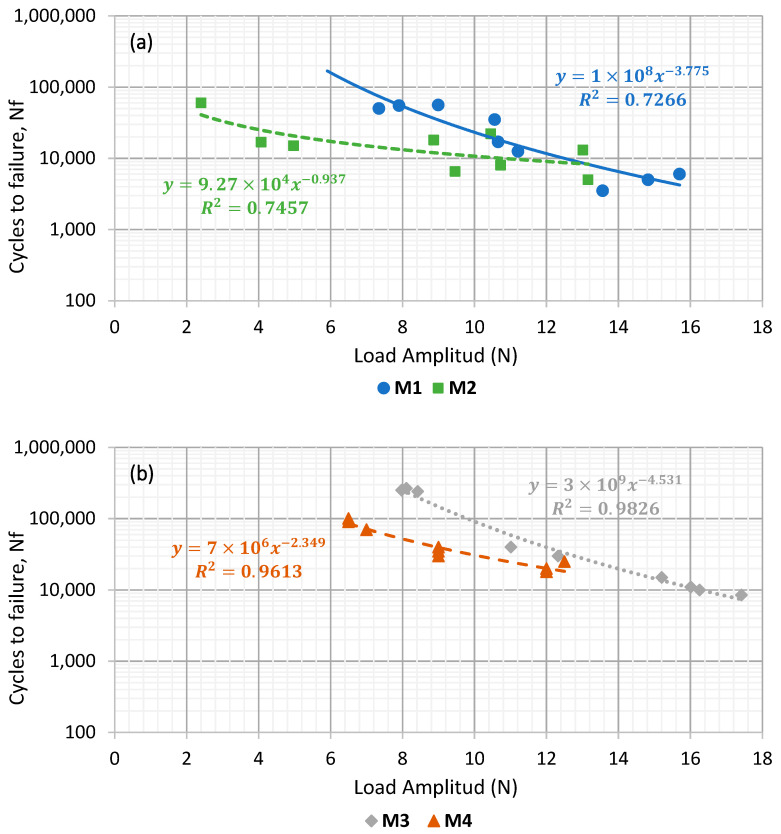
Comparison between types of active filler for mortars manufactured with penetration binder (**a**) and polymer-modified bitumen (**b**).

**Figure 10 materials-15-03307-f010:**
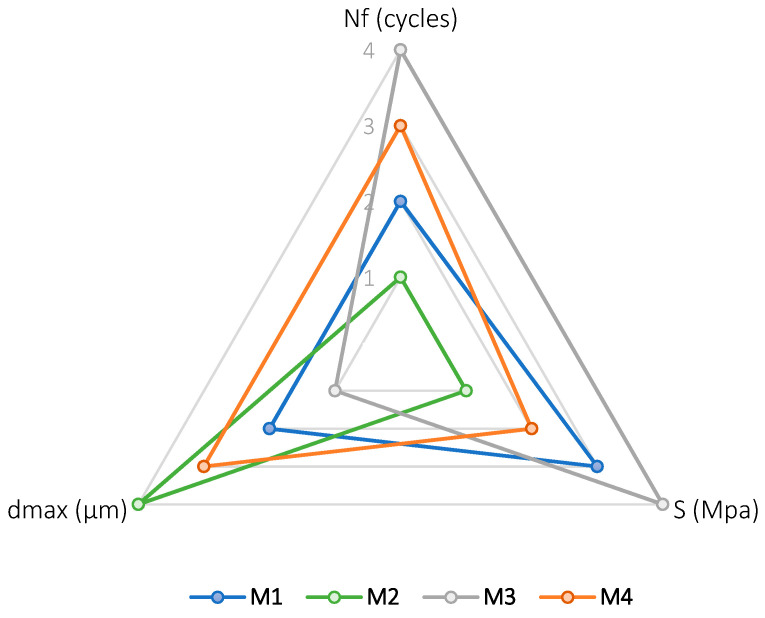
Relationship between maximum deflection, stiffness and fatigue life of the different mortars studied.

**Table 1 materials-15-03307-t001:** Composition of the asphalt mortars studied.

Components	Mortar with Cement + Penetration Bitumen (M1)	Mortar with CaCO_3_ + Penetration Bitumen (M2)	Mortar with Cement + Modified Bitumen (M3)	Mortar with CaCO_3_ + Modified Bitumen (M4)
Asphalt bitumen B35/50 (% over the total mortar weigth)	8.0	8.0	-	-
Asphalt bitumen PMB 25/55–65 (% over the total mortar weigth)	-	-	8.0	8.0
Limestone Sand (% over the total mortar weigth)	64.4	64.4	64.4	64.4
Filler: Portlant Cement (% over the total mortar weigth)	27.6	-	27.6	-
Filler: Calcium carbonate (CaCO_3_) (% over the total mortar weigth)	-	27.6	-	27.6

**Table 2 materials-15-03307-t002:** Characteristics of the sand employed.

Property	Sieve Size (mm)	Percentage of Aggregates Passing (%)
Granulometry (EN 933-1) [21]	*2*	100
	*0.5*	18
	*0.063*	0
Sand Equivalent (EN 933-8) [22]		77.0
Density (kg/m^3^) (EN 1097-6) [23]		2770
Water Absorption (%) (EN 1097-6) [23]		0.88

**Table 3 materials-15-03307-t003:** Characteristics of the fillers employed.

Property	Sieve (mm)	Percentage of Filler Passing (%)	
		*Portland Cement*	*Calcium Carbonate*
Granulometry (EN 933-1)	*2*	100	100
	*0.5*	100	100
	*0.125*	100	100
	*0.063*	96.0	94.0
Density (kg/m^3^) (EN 1097-3, Annex A) [24]		2941	2770

**Table 4 materials-15-03307-t004:** Properties of the binders evaluated in the study.

Property	B 35/50	PMB 25/55–65
Binder Penetration (dmm) (EN 1426) [25]	38	32
Softening point temperature (°C) (EN 1427) [26]	54	65

**Table 5 materials-15-03307-t005:** Density of the asphalt mortars tested.

Components	M1	M2	M3	M4
Apparent Density (kg/m^3^) (EN 12697-6) [28]	2453	2338	2423	2373

**Table 6 materials-15-03307-t006:** Average values obtained at a load level of 10 N employed in the construction of the parametric ranking.

	M1	M2	M3	M4
**Nf Average (cycles)**	21,500	8125	22,750	20,750
**Nf Standard deviation (cycles)**	1905.9	3473.1	1283.1	2986.1
**Nf, Ranking**	**2**	**1**	**4**	**3**
**S Average (Mpa)**	311.3	218.5	353.6	245.1
**S Standard deviation (Mpa)**	16.6	34.2	19.57	8.0
**S, Ranking**	**3**	**1**	**4**	**2**
**dmax Average (µm)**	1200	1850	755	1800
**dmax Standard deviation (µm)**	264.6	768.1	77.2	571.5
**dmax, Ranking**	**2**	**4**	**1**	**3**

## Data Availability

The study did not report any data.

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
