# Peer review of "Influence of Type of Filler and Bitumen on the Mechanical Performance of Asphalt Mortars"

_materials, 2022, doi:10.3390/ma15093307_

Round 1

Author Response

Authors thank the editors and reviewers for the time and effort that they have put into reviewing the manuscript. Their suggestions have enabled us to improve our work. Based on the instructions provided, we uploaded the file of the revised manuscript. Accordingly, we have uploaded a copy of the original manuscript with all the changes highlighted by using the track changes mode in MS Word and color red. Appended to this letter is our point-by-point response to the comments raised by the reviewers.

A file is attached with the response

Reviewer 2 Report

In this paper, an experimental procedure for characterizing the mechanical response of asphalt mortars based on a three-point bending test on small prismatic specimens and applying the Dynamic Mechanical Analyzer (DMA) is proposed. The technique is described through the experimental analysis of four mortar types, considering different fillers and binders.

The presented manuscript seems to be a draft, without links to the figures and to the references, with no lines numbering and with highlights and a grey zone on the right side of the pages.

It can be said that the general organization of the manuscript is correct. Nevertheless, some important data and information are missing.  

If the authors agree to consider the comments and suggestions addressed below, a revised version of the paper could be considered for publication in MDPI Materials Journal.

  • Abstract: In line 4, it says “..diferentes…”
  • Introduction: It is mentioned that there are several tests for evaluating mortar performance. However, no other method is described.
  • Materials: Height of first cylindrical samples is not informed.
  • Testing plan: Some information is missing: layout of the three point bending tests, description of the used equipment, strain amplitudes, etc. Regarding that those issues are the basis of the proposed method, further information should be provided.
  • Results: Authors do not include a precise description of each figure. Further details should be included in the final version of the paper.
  • Considering that a new methodology is proposed, a comparison with other available techniques should be included in the paper.

Author Response

Authors thank the editors and reviewers for the time and effort that they have put into reviewing the manuscript. Their suggestions have enabled us to improve our work. Based on the instructions provided, we uploaded the file of the revised manuscript. Accordingly, we have uploaded a copy of the original manuscript with all the changes highlighted by using the track changes mode in MS Word and color red. Appended to this letter is our point-by-point response to the comments raised by the reviewers.

A file with the response is attached

Reviewer 3 Report

Dear Authors and Editors,

Here are some comments concerning the reviewed manuscript:

  1. The manuscript style is somewhere wordy and should be corrected for grammatic and stylistic errors. 
  2. The CaCO3 formula should be written with lowerscript CaCO3
  3. Mg/mshould be rather written as mg/mas M stands for mega not for milli
  4. Why in fig. 7 every composition M1-M4 has different number of measurements (dots)?
  5. Fig. 8 has also different type of points and the do not match with each other and with the same color in fig. 7. Could the authors somehow comment this experimental design details?
  6. What is the difference between fig. 8 and 9? Seems that they have similar legend and correlation models

Author Response

Authors thank the editors and reviewers for the time and effort that they have put into reviewing the manuscript. Their suggestions have enabled us to improve our work. Based on the instructions provided, we uploaded the file of the revised manuscript. Accordingly, we have uploaded a copy of the original manuscript with all the changes highlighted by using the track changes mode in MS Word and color red. Appended to this letter is our point-by-point response to the comments raised by the reviewers.

A file with the responses is attached

Reviewer 4 Report

The authors carried out a detailed study of the influence of the asphalt mortar composition on the mechanical properties and the interaction between filler and biding, using DMA.

Overall, the work is well written, the introduction reports the theoretical background associated with the topic of the article, the figures are beautiful and didactic, the methodology is well described, with some details that need to be clarified. The discussion relates the results to the objectives and previous literature. However, some points need to be clarified before publication:

1) Clarify what is the innovation, that is, what is not yet known about the topic addressed: a) Because in the abstract the authors report that the use of DMA to characterize the mechanical properties of mortars is new. However, DMA is a well-known technique. If there is innovation in this sense, it needs to be clarified. b) In Testing Plain: “This investigation has the main objective of studying the effect of the mastic components in the mechanical performance and fatigue life of asphalt mortars manufactured with different types of active filler and asphalt binder”. The components are already known, what information is missing about them?

2) It is necessary to clarify the meaning of the points presented in figures 6, 7, 8 because it seems that does not correspond to what is described in the Testing plan: “Based on these considerations, for each type of mortar 9 specimens were tested divided into 3 sub-groups of 3 specimens. Each sub-group was tested until fatigue cracking failure at 3 different strain amplitudes (strain-controlled mode) by applying a sinusoidal load at a frequency of 5 Hz and at a temperature of 35 ⁰C.” Furthermore, it seems to me that some specimen compositions were more tested than others. What is the reason for this difference?

3) Usually the results of the “groups” or “sub-groups” are represented by mean and standard deviation (when data are parametric). And the difference, reported by statistical methods. For example Two-way ANOVA can be used to analyze mortar components simultaneously. In this way, I believe that there are improvements that must be made in the presentation of the results and in their analysis, before the publication of the article.

4) Figure 10 is beautiful, but I didn't understand the values ​​with zero cycles. In addition, if each point represents the mean, it must be accompanied by the standard deviation.

Minor comments:

Correct typos such as CaCo3 (the 3 is subscript), Equation and no ecuation, remove the highlighted parts

Author Response

Authors thank the editors and reviewers for the time and effort that they have put into reviewing the manuscript. Their suggestions have enabled us to improve our work. Based on the instructions provided, we uploaded the file of the revised manuscript. Accordingly, we have uploaded a copy of the original manuscript with all the changes highlighted by using the track changes mode in MS Word and color red. Appended to this letter is our point-by-point response to the comments raised by the reviewers.

I file with the response is attached

Round 2

Author Response

Authors thanks for the review.

The final version of the manuscript addressed the suggestion made by the reviewer.

Reviewer 2 Report

Authors have considered all comments and suggestions. The paper is ready to be published.

Author Response

(The authors gave the same response as above.)
